# Organic Anion Transporters (OAT) and Other SLC22 Transporters in Progression of Renal Cell Carcinoma

**DOI:** 10.3390/cancers14194772

**Published:** 2022-09-29

**Authors:** Thomas C. Whisenant, Sanjay K. Nigam

**Affiliations:** 1Center for Computational Biology and Bioinformatics, University of California, San Diego, CA 92093-0693, USA; 2Department of Pediatrics, University of California, San Diego, CA 92093-0693, USA; 3Department of Medicine, University of California, San Diego, CA 92093-0693, USA

**Keywords:** kidney cancer, remote sensing and signaling, SLC22, KIRC, KIRP, SLC22A1, SLC22A15, SLC22A18, SLC22A23, SLC22A24, SLC22A5, SLC22A4, the Cancer Genome Atlas

## Abstract

**Simple Summary:**

Kidney cancer diagnoses make up over 2% of newly identified cancers each year. SoLute Carrier 22 (SLC22) genes are expressed in normal functioning kidneys and are responsible for transport of myriad metabolites, xenobiotics, antioxidants and other small molecules, but their role in kidney cancer is not well understood. We assessed the relationship between the expression of SLC22 genes and survival in patients with kidney cancer and found expression patterns of multiple SLC22 genes to be associated with overall survival as well as with disease progression. We sought to interpret this data in the context of physiological information from previous studies by us and others. Furthermore, network analysis indicated the importance of SLC22 genes and identified additional genes that might be therapeutic targets.

**Abstract:**

(1) Background: Many transporters of the SLC22 family (e.g., OAT1, OAT3, OCT2, URAT1, and OCTN2) are highly expressed in the kidney. They transport drugs, metabolites, signaling molecules, antioxidants, nutrients, and gut microbiome products. According to the Remote Sensing and Signaling Theory, SLC22 transporters play a critical role in small molecule communication between organelles, cells and organs as well as between the body and the gut microbiome. This raises the question about the potential role of SLC22 transporters in cancer biology and treatment. (2) Results: In two renal cell carcinoma RNA-seq datasets found in TCGA, KIRC and KIRP, there were multiple differentially expressed (DE) SLC22 transporter genes compared to normal kidney. These included SLC22A6, SLC22A7, SLC22A8, SLC22A12, and SLC22A13. The patients with disease had an association between overall survival and expression for most of these DE genes. In KIRC, the stratification of patient data by pathological tumor characteristics revealed the importance of SLC22A2, SLC22A6, and SLC22A12 in disease progression. Interaction networks combining the SLC22 with ADME genes supported the centrality of SLC22 transporters and other transporters (ABCG2, SLC47A1) in disease progression. (3) Implications: The fact that many of these genes are uric acid transporters is interesting because altered uric acid levels have been associated with kidney cancer. Moreover, these genes play key roles in processing metabolites and chemotherapeutic compounds, thus making them potential therapeutic targets. Finally, our analyses raise the possibility that current approaches may undertreat certain kidney cancer patients with low SLC22 expression and only localized disease while possibly overtreating more advanced disease in patients with higher SLC22 expression. Clinical studies are needed to investigate these possibilities.

## 1. Introduction

SoLute Carrier (SLC) genes are translated into membrane bound proteins critical to the transport of small molecules, including a wide range of xenobiotics and metabolites, in and out of organs [1]. For example, the gut-liver-kidney axis is critical for homeostasis as well as drug absorption, distribution, metabolism, and excretion (ADME) [2,3,4]. There are currently 66 human subfamilies of the SLC family, encompassing over 450 uniquely defined genes. These include multi-specific, oligo-specific and monospecific transporters of solutes, among which are many endogenous small organic molecules of considerable importance in physiology and pathophysiological states [5].

One SLC family that has received a great deal of attention is SLC22 [6,7]. This family, first identified in 1997 with three members (SLC22A6 or NKT or OAT1, SLC22A7 NLT or OAT2, and SLC22A1 or OCT1), now consists of over 30 members and is perhaps best known for its multi-specific “drug” transporters (e.g., organic cation transporters and organic anion transporters) [8]. SLC22 transporters have, in recent years been shown to play an essential role in endogenous metabolic processes [9,10,11,12]. This includes the regulation of a variety of important small molecules such as vitamins, uric acid, gut microbe-derived products, antioxidants, nutrients [13,14,15].

Originally, the SLC22 family was divided into three clades: organic cation transporters (OCTs) and organic anion transporters (OATs), and organic zwitterion/cation transporters (OCTNs) and additionally divided into subclades based on phylogenetic analysis [16]. More recently, these groupings have been further refined based on structural and genomic homology, metabolite interactions, and network characteristics based on substrate specificity, revealing 8 distinct subgroups [17]. While the role of SLC22 transporters in metabolic disease and chronic kidney disease is now apparent [18], the role of SLC22 transporters in other pathophysiological contexts is just beginning to be explored. In contrast to ABC transporters (e.g., p-glycoprotein, ABCG2 or BCRP, and members of the MRP or ABCC family), which transport drugs and have been studied in the context of cancer and tumor resistance, much less is known about the role of SLC22 “drug” transporters (and other SLC22 family members) in cancer.

Previous work showing a link between increased SLC gene expression and cancer has focused on the upregulation of glucose, amino acid, and lactate transporters, presumably in order to meet the increased nutrient and waste removal demands of cancer cells [19]. More recently, certain SLC22 family genes have been implicated as possible tumor suppressors, as in the cases of SLC22A7 in Hepatocellular Carcinoma [20] and SLC22A1 in Cholangiocellular Carcinoma [21]. Another suggested role for SLC22 is in facilitating uptake of anticancer drugs. For example, an observed connection between increased SLC22A1, SLC22A2, and SLC22A3 expression and positive response to chemotherapy in colon cancer patients is thought to be due to the role of these genes specifically transporting oxaliplatin into cancer cells [22,23,24].

Globally, renal cancer represented 2.2% of new cancer cases in 2020, totaling over 431,000 instances [19]. Major risk factors include smoking and obesity [25] and there is a higher prevalence in men that has been attributed to differential rates of smoking between men and women. That said, there are higher obesity rates in women diagnosed with renal cancer [26]. Interestingly in women, the mortality rate is lower than the rate of diagnosis, which has led to the suggestion of a an “obesity paradox” which presumes a survival advantage in obese individuals diagnosed with renal cancer compared to non-obese people [27]. Other risk factors include hypertension, chronic kidney disease/end stage renal disease, environmental exposure to carcinogenic chemicals, as well as inheritance of genetic alterations (insertions, deletions, mutations) that predispose individuals to development of kidney cancer [28].

There are 3 major forms of renal cell cancer, representing 90% of all cases. These are: clear cell Renal Cell Carcinoma (ccRCC, 75%), papillary cell (PRCC, 10%), and chromophobe (ChRCC, 5%). ccRCC is twice as prevalent in men compared to women, and PRCC is more than 3:1 as prevalent in men; in contrast, ChRCC is somewhat more common in women who have ~55% of all cases of this form [29]. Since 1977, 5-year overall survival rates have improved considerably. There is significant stratification by stage at diagnosis, with only 12% 5-year survival for individuals with metastatic (Stage IV) disease [30].

Here, we used datasets available from The Cancer Genome Analysis (TCGA) project to explore the role of SLC22 transporters in clear cell renal carcinoma (KIRC) and papillary cell Renal Cell Carcinoma (KIRP). The analysis revealed an association between overall survival and expression of many SLC22 genes between tumor and normal tissue. Patient data was stratified according to tumor stage, tumor size and progression, lymph node involvement, and presence of metastasis, providing insight into the potential role of particular SLC22 transporters and their interactions with other genes, metabolites, and xenobiotics in disease progression.

## 2. Materials and Methods

### 2.1. Statistical Analysis

All analyses, unless otherwise described, were performed in the R statistical computing environment (v3.5.1) along with the standard functions from the bioconductor packages for biostatistics functions. The packages used for specific analyses are described below.

### 2.2. TCGA RNA-Seq Analysis

Comparison of specific tumor type and matched normal tissue gene expression data for SLC22 gene was performed with the GEPIA online graphical user interface, (http://gepia2.cancer-pku.cn, accessed on 19 May 2021, [31]). This tool was used to produce gene specific expression boxplots for KIRC and KIRP tumor along with TCGA and GTEX normal kidney sample data. Within the boxplots, significant differences between the tumor and normal samples were determined as absolute log fold change (LogFC) greater than 1 and q-values less than 0.01 based on pre-loaded limma analyses.

KIRC and KIRP normalized RSEM estimate count data and associated metadata were obtained from the TCGA portal (http://portal.gdc.cancer.gov, accessed on 22 May 2021). The subset of expressed genes in each cancer were defined as having greater than 75 counts in at least 50% of the tumor samples. Differential expression (DE) analysis along with normalization and generation of log-transformed counts per million (LogCPM) were performed with the R packages limma [32], and specific application of the limma-voom algorithm [33]. Limma models included metadata factors age, gender, and reported race as determined in the survival analysis for each gene. A threshold of 0.05 for the Benjamini-Hochberg adjusted *p*-value was used to determine the significantly different genes [34]. All expression plots were created with the R package ggpubr (v 0.2.5).

### 2.3. Survival Analysis

Initial overall survival analysis results were obtained with the OncoLnc online graphical user interface (www.oncolnc.org, accessed on 3 May 2021, [35]). This tool generates, for each expressed gene in each cancer type, a Cox proportional hazards model for overall survival fitted using the gene expression values and three metadata variables: AJCC Pathologic Stage, gender, and age. The output for each variable in the model is a Cox coefficient and an associated *p*-value that can be used to determine if the relationship between overall duration of patient survival and the variable are significant. Negative Cox coefficients correspond to variables where higher values (and higher expression for genes) correlate with longer survival; and conversely for positive Cox coefficients. Genes were identified as significantly associated with a cancer type if the adjusted *p*-value was less than 0.05. For the significant genes, the OncoLnc tools generates Kaplan–Meier survival analysis curves based on the sets of patients with the highest (top tercile) and lowest (bottom tercile) expression of the gene. Logrank test *p*-values reporting the significance of the difference between the groups are also reported with the plot.

Additional survival analysis performed with functions from the R package Survminer (0.4.6). Using the available data from TCGA, the primary endpoint used for this analysis was overall duration of patient survival measured in days post-diagnosis of disease. The Cox proportional hazards models used in these analyses included the age and gender from the “base” model in the OncoLnc analysis and additional metadata variable combinations (Appendix A) using the clinical metadata from the TCGA portal (Appendix A). AJCC Pathologic Stage was not used in all models as AJCC Pathologic Stage depends upon some of the variables we were investigating (i.e., tumor size and progression, presence of metastasis). The threshold for significance of the *p*-values associated with the analysis of each Cox proportional hazards model was determined by dividing 0.05 by the total number of tested models in each cancer (0.05/221 = 2.2 × 10^−4^). For each significant SLC22 gene, the model with the lowest *p*-value was chosen and reported along with the associated Cox coefficient and Hazard Ratio (Table 1). The directionality and magnitude of these values, while consistent with the OncoLnc results, are often greater, suggesting that the added variables increase the variance assignable to known factors and better isolate the impact of the expression of each SLC22 gene on overall survival.

These models were further analyzed in order to test whether the inclusion of the metadata variables violated the assumption of proportionality. This is determined by implementing a chi-square test comparing the full (all variables) proportional hazards model with the null model using no variables. Models with *p*-values less than 0.01 generated by this test indicate a violation of the proportionality assumption and were not further investigated (Appendix A).

Each significant model was analyzed with the gene expression variable removed. A Hazard Ratio table for a Cox proportional hazards model incorporating age, gender, reported race, and treatment type in KIRC produces a significant *p*-value for the age variable (Appendix A). This variable was part of every model tested (including the OncoLnc models) and thus no further analysis was performed. For the model associated with SLC22A7 in KIRC (Appendix A), age was also significant while the two KIRP models had no variables that were significant (Appendix A). These data show that, with the exception of age in KIRC, none of the metadata variables used in the combinations shown can stratify patients by their duration of overall survival. These graphical summaries were generated with survminer’s ggforest function (v0.4.6).

### 2.4. Network Analysis

Interactions between DE genes for each comparison were identified using the STRING database of protein–protein interactions. The STRING medium confidence (Combined Score > 0.4) interactome is an assembly of physical and functional interactions, biological knowledge, and various computational metrics that include data mining and homology modeling that are used as input to generate a “Combined Score” [36]. These interactions were visualized within Cytoscape (v3.8.0) [37] with singletons removed. Networks were annotated with data from the log fold change from the DE analysis and genes in the top ~15% based on degree were noted.

## 3. Results

### 3.1. Decreased Expression of SLC22 Transporters in Kidney Cancers and Association with Poorer Outcomes

Of the human SLC22 transporters of organic cations, anions, and zwitterions, there are 17 expressed at detectable levels in KIRC and KIRP based on the publicly available RNA-Seq data from The Cancer Genome Atlas (TCGA, http://portal.gdc.cancer.gov, accessed on 24 March 2021). Using the GEPIA2 tool (http://gepia2.cancer-pku.cn, accessed on 19 May 2021, [31]), we analyzed expression differences in a large cohort of primary tumors and normal kidney tissue and identified 6 genes significantly DE in both KIRC and KIRP (Figure 1A). Five of those are found in comparisons of both tumor types: SLC22A6/OAT1 (Figure 1B), SLC22A7/OAT2 (Figure 1C), SLC22A8/OAT3 (Figure 1D), SLC22A12/URAT1 (Figure 1E), SLC22A13/ORCTL3 (Figure 1F). For each of these DE genes, expression was decreased in the primary tumor sample groups, and the difference was significant based on limma differential gene expression analysis (adjusted *p*-value < 0.01) [32].

The categorizations of family and clade are shown for each of the SLC22 genes expressed in the kidney (Table 2). In the updated grouping, OATs are considered to be organic anion transporters, OCTs are considered organic cation transporters, and OCTNs are considered zwitterionic transporters. However, the associations implied by these general groupings are not absolute, especially for the groupings designated “-related” [17]. Interestingly, of the DE genes between tumor and normal kidney tissue (in bold), SLC22A6, SLC22A7, and SLC22A8 are OAT1, OAT2, and OAT3. These are genes that function in the kidney to mediate transfer of a wide variety of organic anion molecules (including endogenous and xenobiotic molecules) from the blood to the proximal tubule via the basolateral membrane [14]. SLC22A12 (URAT1) and SLC22A13 are found on the apical membrane of the kidney and transport urate, amongst other substrates [38,39].

To further investigate the observation that SLC22 gene expression changes upon presentation of primary tumors in the kidney, we looked at overall survival times for patients suffering from KIRC and KIRP in the context of SLC22 gene expression (Appendix A, Column J). In the kidney, SLC22 transporters are considered excellent markers of differentiation [44,45,46]. Thus, the hypothesis was that decreases in SLC22 gene expression correlate with (and possibly contribute to) reduced duration of overall survival following initial diagnosis in these tumor types. We used our list of expressed SLC22 genes as input into the OncoLnc tool [35], which searches for significant correlations between gene expression and overall survival in 12 different cancer RNA-Seq datasets available from the TCGA Project. The outputs are Cox coefficients, raw, and adjusted *p*-values for all expressed SLC22 genes in the KIRC and KIRP datasets (Table 1). Negative Cox coefficients correspond to variables where higher values (and higher expression for genes) correlate with longer survival and conversely for positive Cox coefficients. A majority of the SLC22 genes (12 of 17) demonstrate a significant association (B-H adjusted *p*-value < 0.05) with overall survival in KIRC while only 4 genes [SLC22A2, SLC22A13, SLC22A18, and SLC22A24] are significantly associated in KIRP.

Of the significantly DE genes between KIRC primary tumors and normal kidney tissue, SLC22A6 (Figure 2A, [41]), A7 (Figure 2B), A8 (Figure 2C), A12 (Appendix A, [43]), and A13 (Figure 2D) also showed significant association with overall survival. Although the first three genes are multi-specific (SLC22A6, SLC22A7 and SLC22A8) and the latter two (SLC22A12 and SLC22A13) are relatively monospecific, it is worth noting that all of them are strongly implicated in transport of the anti-oxidant, uric acid. This may be of particular clinical-translational interest since, in a prospective analysis of the UK Biobank, high serum uric acid (SUA) levels were associated with higher incidence of renal cancer [47]. Other studies have also associated uric acid levels with renal cancer and outcomes [48,49]. For each of these genes, the signs of the Cox coefficient were all the same direction (negative). SLC22A13 was also significantly associated with overall survival in KIRP and had a negative Cox coefficient (Appendix A). Two other genes, SLC22A2 (OCT2, Appendix A) and SLC22A24 (Figure 2E and Appendix A) were significant in both KIRC and KIRP and had negative Cox coefficients. Consistent with our hypothesis, the negative Cox coefficients for these genes indicate that higher gene expression is observed in samples from patients with longer survival times. These results also extend the previously observed trend of decreased expression in tumor samples by showing that SLC22 expression further decreases in patients with poorer outcomes.

### 3.2. Variables Other Than OAT Family Expression Associated with Decreased Survival

Because these are transporters of endogenous metabolites and drugs—and, in the case of SLC22A6, SLC22A8, SLC22A12, established drug targets—these observations are of interest from both the viewpoint of cancer biology and therapeutics [50]. Therefore, we further investigated the relationship between overall survival and gene expression. To expand on the previous OncoLnc Cox proportional hazards model analysis using the metadata variables age and gender, we included additional demographic and oncologic variables available from the TCGA (Appendix A). These variables were: reported race, type of treatment (Radiation or Pharmaceutical), prior malignancy, tumor size and progression (AJCC Pathologic T), lymph node involvement (AJCC Pathologic N), and presence of metastasis (AJCC Pathologic M). After analyzing all variable combinations (Appendix A), there were 9 of 12 (SLC22A2, SLC22A4, SLC22A5, SLC22A6, SLC22A7, SLC22A8, SLC22A11, SLC22A12, SLC22A24) (Appendix A) and 4 of 4 (SLC22A2, SLC22A13, SLC22A18, SLC22A24) (Appendix A) SLC22 genes still significantly associated with overall survival in KIRC and KIRP, respectively. From these results, we determined that the oncologic variables tested were not part of highest performing proportional hazards models and a separate analysis utilizing these variables in limma models for differential expression is required to determine their relationship to SLC22 gene expression.

### 3.3. Influence of OAT (and Other SLC22 Transporter) Expression on Tumor Stage and Individual TNM Parameters in the Context of Overall Survival

The survival analysis tracks expression in the tumor with long term patient outcomes and suggests previously unidentified roles of OATs and other SLC22 genes in cancer biology and clinical outcomes. Since there are accepted criteria for the staging of malignant solid tumors for the purposes of estimating both disease progression, as well as the overall duration of intervention-less survival, we hypothesized that there is a relationship between the aforementioned SLC22 genes and the criteria for stage determination, specifically: tumor size and progression (AJCC Pathologic T), lymph node involvement (AJCC Pathologic N), and presence of distant metastasis (AJCC Pathologic M). To focus specifically on the relationship between our genes of interest and these pathologic classifications, the TCGA clinical metadata variables (age and gender) were added to the models used in DE comparisons. We determined there is no direct correlation between SLC22 gene expression and age or gender in the TCGA renal cancer data. In addition, there is a precedent for using these variables in linear models to obtain lists of differentially expressed genes [31]. Furthermore, in two separate Cox proportional hazards models in KIRC that were independent of gene expression, age was significantly associated with overall survival (Appendix A). These results suggest that any biological variation associated with age will be removed in the limma models and increase our confidence in any significant DE SLC22 genes. Additionally, based on the overall survival analysis data discussed above, reported race was included in the limma models. For both KIRC and KIRP, comparisons were made across each of the oncologic variables for the subset of SLC22 genes where sufficient samples had detectable gene expression levels.

While multiple comparisons had no DE genes in the entire expressed gene list (i.e., Stage III vs. Stage IV), there were 6 DE SLC22 genes in the Tumor Size and Progression comparison in KIRC [SLC22A2, SLC22A5, SLC22A6, SLC22A11, SLC22A12, SLC22A18] (T3 v T4, Figure 3A, Table 3 and Appendix A). The comparison of absence vs. presence of Metastasis (M0 v M1, Figure 3B) revealed 4 DE SLC22 genes (SLC22A2, SLC22A6, SLC22A12, SLC22A23). SLC22A2, SLC22A6, and SLC22A12 were DE in both these comparisons (T3 v T4 and M0 v M1); moreover, SLC22A2 was also DE in the comparison of positive versus negative lymph node involvement (N0 v N1, Figure 3C). Two other comparisons, Stage I vs. II and Stage II vs. III, displayed significant differences in SLC22A17 expression (Figure 3D).

In KIRP, only positive versus negative lymph node involvement (N0 vs. N1) showed any DE SLC22 genes which were SLC22A2, SLC22A18, SLC22A3, SLC22A11, and SLC22A5 (Appendix A). In comparing the lists of DE genes from KIRC and KIRP associated with these oncologic staging criteria to those that were either DE between tumor versus normal tissue or implicated in overall survival (highlighted in orange in Appendix A), these results support the conclusion that some SLC22 genes are important throughout development and progression of kidney cancer.

### 3.4. Biological Support and Network Interpretation of the Role of SLC22 Genes in KIRC and KIRP Disease Progression (in the Context of the Remote Sensing and Signaling Theory)

SLC22-interacting genes are known to play a role in absorption, distribution, metabolism, and excretion of endogenous small molecules as well as drugs and toxins. For example, an interaction network of 690 ADME-related and other genes revealed SLC22 genes to be central within the network structure [12]. It was shown that there exists a highly connected network of ADME and other genes that is co-expressed across the gut, liver, and kidney tissues that likely plays a role in system level metabolism of xenobiotics (including chemotherapy drugs) and endogenous small molecules and metabolites. According to the Remote Sensing and Signaling Theory, such a network is critical for inter-organ (e.g., gut-liver-kidney) and inter-organismal (e.g., gut microbe-host) communication via endogenous small molecules [1,6,15]. More generally, the Remote Sensing and Signaling Theory addresses the roles of multi-, oligo- and monospecific transporters in small molecule communication across organelles, cells, organs, organ systems, and organisms. The SLC22 transporter family is particularly noteworthy in that it consists of multi-, oligo,- and monospecific transporters with established in vivo and in vitro roles in the regulation of metabolites, signaling molecules, antioxidants, nutrients, and gut microbe products. It is possible that an accumulation of perturbations in genes of the SLC22 transporter family and/or other genes in the larger Remote Sensing and Signaling network, driven by the gene expression changes associated with KIRC and KIRP disease progression, contributes to reduced overall survival for these patients.

Similarly, SLC22 interactions with other compounds including metabolites, signaling molecules, antioxidants, nutrients and chemotherapeutic drugs are likely to contribute to overall patient survival. Given the reduced expression observed for many SLC22 genes, the downstream effect might be a shift in the SLC22 transporter-mediated uptake of these small molecules into tumor cells as the disease progresses across the multiple pathologic stages. Identification of the relevant interacting compounds for each SLC22 gene requires a multifaceted approach. Based on our work and others, we have compiled the compounds that have been associated with the SLC22 transporters discussed here (Appendix A). The data comes from in vitro transport assays, in vivo knockout mouse metabolomics, and GWAS studies. The asterisks indicate that metabolite data for SLC22A6 and SLC22A8 comes from papers published by the senior author’s group [4,10,16,51,52,53,54,55,56,57,58]. The non-drug data is partly adapted from supplementary information in Engelhart et al., 2020 [17] with the addition of some other published data from the senior author’s group [59,60]. The data on the association of these transporters with endogenous metabolites, signaling molecules, nutrients, and antioxidants may shed mechanistic light on tumor growth, invasion and/or metastasis. The chemotherapeutic drug data was found in the UCSF-FDA Transportal [61] where there is information on SLC22A2, SLC22A5, SLC22A6, and SLC22A8 interactions with chemotherapeutic drugs. Loss of these uptake transporters with disease progression could conceivably play a role in tumor resistance to certain chemotherapeutics.

Of the 690 ADME-related and other genes in the aforementioned network, 430 were expressed in KIRC, and 511 were expressed in KIRP. For these expressed network genes, we performed the same comparisons for differential expression across the T, N, and M categories described above for the SLC22 genes (Appendix A). The DE genes were combined with the DE SLC22 genes to generate networks and calculate network metrics using the STRING interaction database [36]. The two largest networks were generated using the DE genes from the tumor size/progression class, T3 v T4 (156 nodes, Figure 4A), and presence of metastasis, M0 v M1 (158 nodes, Appendix A), comparisons. Analysis of each network was performed to generate a series of metrics including degree, betweenness centrality, and closeness centrality (Appendix A). Consistent with our hypothesis that the SLC22 family plays an important role in the progression of KIRC, as measured by degree, four SLC22 genes are found in the top 10 most connected genes in the T3 v T4 network (Figure 4B).

Two interesting ADME genes that were ranked in the top 5 by closeness centrality in each of the KIRC networks are ABCG2 and SLC47A1. In addition to being central to each of these networks, they are DE in each comparison. Violin plots show the changes in these two genes for tumor size/progression, T3 vs. T4 (Figure 5A), lymph node involvement, N0 v N1 (Figure 5B), and presence of metastasis, M0 v M1 (Figure 5C). Furthermore, SLC47A1 is DE in the KIRP comparison of lymph node involvement classes, N0 v N1 (Figure 5D).

As seen with the SLC22A2 and SLC22A12 genes, expression changes across oncologic variables can be accompanied by a correlation with overall survival. Using the OncoLnc tool, we observed that both ABCG2 and SLC47A1 had significant and negative Cox coefficients, −0.36 (adjusted *p*-value = 3.0 × 10^−4^) and −0.415 (2.2 × 10^−5^), respectively, indicating that decreased expression of these genes is associated with reduced overall survival duration (Figure 6A,B). Interestingly, while SLC47A1 shows a significant decrease in expression in KIRP N0 vs. N1, its expression is not significantly associated with overall survival in these patients. These data indicate that the central genes in the network of transport and metabolic factors that are changing in KIRC are also important to the overall survival of patients with the disease.

## 4. Discussion

Many SLC22 transporters have high and sometimes nearly exclusive expression in the kidney [12,38]. Downregulation of various SLC22 subfamily genes have been associated with kidney disease [62]. For example, the SLC22 genes SLC22A6/OAT1 and SLC22A8/OAT3 participate in the transport of uremic toxins during renal excretion, and accumulation of certain uremic toxins can contribute to the progression of chronic kidney disease [6,55]. Specifically, we observed large reductions in expression of multiple SLC22 genes in two of the most common types of kidney cancer, ccRCC/KIRC and PRCC/KIRP (Figure 1A). Multiple OAT and OAT-like genes were downregulated in both cancer types. When looking at the expression patterns of these genes within the normal and tumor sample populations, it is clear that while these changes are significant, there is a large population without decreased expression in their tumor and also a number of individuals without high SLC22 expression in their normal kidneys (Figure 1B). For those individuals with decreased SLC22 gene expression, there is a strong potential for systemic complications arising from the reduced capacity to metabolize the uremic toxins and other small molecules.

Before discussing the SLC22 genes identified in this study in relation to renal cell cancer and various oncologic variables (TNM class) used to determine pathologic stage, we note previous work has shown that SLC22A2, SLC22A12, and SLC22A13 are downregulated in human kidney tumors [43,56,63]. Based on these observations and the availability of the OncoLnc tool [35], we investigated the connection between SLC22 gene expression and overall survival in patients with KIRC and KIRP primary tumors. In addition to confirming previously published results showing an association between overall survival and expression of SLC22A1/OCT1 and SLC22A2/OCT2, SLC22A5/OCTN2, SLC22A6/OAT1, and SLC22A12/URAT1 in KIRC [41,43,64,65], multiple additional SLC22 genes of the OAT and OAT-related subclades were also associated with overall survival, including SLC22A7, SLC22A8, and SLC22A13 (Figure 2 and Appendix A). These genes were also DE between KIRC tumors and normal kidney. In KIRP, there were three genes associated with overall survival that were also significant in KIRC: SLC22A2, SLC22A13, SLC22A24. The patients with lower expression of each of these SLC22 genes had shorter overall survival durations, raising the possibility that reduced capacity for transport by these transporters has systemic effects on the individual.

With this overall survival information in hand, we performed a deeper investigation into the oncologic variables (TNM classification) used to classify tumor behavior as well as interacting genes. This resulted in novel insights likely to be of interest in the contexts of cancer biology and therapeutics. The base Cox proportional hazards model used by the OncoLnc tool incorporates the additional variables of age, gender, and stage (determined by pathology) in order to better account for the relationship between each gene’s expression and duration of overall survival. To further investigate this relationship, we sequentially incorporated additional demographic and tumor associated variables into Cox models. The demographic variable of reported race was part of nearly all the top scoring models for SLC22 genes in both KIRC and KIRP, and the *p*-values associated with these models were lower than the original base models (Table 2). While inclusion of this variable can be problematic due to nuances associated with collecting this information (i.e., underrepresentation of the Asian subgroup), the data suggest that, in the context of specific SLC22 gene expression, it can improve our ability to identify those patients who might need more aggressive treatment due to reduced overall survival times.

Another variable identified in multiple highest performing models in KIRC is “treatment type.” Given that this variable was a significant covariate in most of the KIRC models, we took it as an indication that something about those treatment groups increases the variance in gene expression values within other variables of interest to us. This variable, treatment type, has two possible values: Pharmaceutical and Radiation. The variable was not present in any of the models that violated the assumption of proportionality, and it was not significant in the proportional hazards model without gene expression values. The standard of care for KIRC and KIRP, following surgical resection of the tumor, varies depending on the staging information—where more localized disease (Stage I) will most likely be treated through radiation while advanced (or at least non-localized) disease will be treated with pharmaceuticals or a combination of both [66]. Interestingly, there is strong evidence that patients receiving blood transfusions at the time of surgery had significantly increased disease recurrence and cancer specific mortality [67]. With both the observation that surgical details are important and that accounting for treatment type in the model improves the association of SLC22 gene expression and overall survival, it is conceivable that the current criteria used for deciding the course of treatment may be undertreating localized disease when SLC22 gene expression is low or overtreating more progressive disease where SLC22 gene expression is still elevated. It is also undergirding the need for biomarkers that can be used to personalize treatment regimens, where expression of specific SLC22 genes could be part of the panel for KIRC and KIRP.

An interesting observation in the deeper survival analysis is the lack of the oncologic variables related to disease progression (tumor size and progression, lymph node involvement, and presence of metastasis) in the best performing models. This suggests that SLC22 gene expression correlates with these variables; and it has been established that these variables contribute to assignment of stage and stage correlates with overall survival [40]. On the other hand, the potential correlation with oncologic variables made it likely that SLC22 gene expression was significantly changed across the various levels of the oncologic variables. Differential expression analysis did identify many SLC22 genes that were changed in KIRC but only a handful in KIRP (Table 3). Some of these genes were found in multiple comparisons. In KIRC, the SLC22 genes significant in the most comparisons analyzing disease progression were SLC22A2, SLC22A6, and SLC22A12. Furthermore, SLC22A2 was seen to be DE between patients with and without lymph node involvement in both KIRC and KIRP.

SLC22A12 is DE between KIRC tumors and normal kidney, associated with overall survival in KIRC and DE between tumor size and progression (T3 vs. T4, Figure 3A), stage I v II, and presence of metastasis (M0 vs. M1, Figure 3B). It has been identified as a biomarker of KIRC and shown to inhibit cancer cell growth in vitro when overexpressed [43]. SLC22A12 (aka URAT1) is, additionally, best known as a kidney transporter of urate, an antioxidant, which may be relevant to tumor behavior. Indeed, many of the OATs and OAT-related SLC22 transporters identified here are known urate transporters (SLC22A6, SLC22A7, SLC22A8, SLC22A11, SLC22A12, SLC22A13) [42,57,58]. In the context of treating uric acid disorders, several of these transporters are established drug targets (SLC22A6, SLC22A8, SLC22A12). The effects of these existing drugs on kidney cancer biology is worthy of further investigation.

As a first step toward a biological and/or clinical interpretation of the observed SLC22 gene expression changes, we curated a list of endogenous small molecules and chemotherapeutic drugs that have documented interaction with SLC22 transporters. These interactions, compiled from our own prior studies and available databases containing the work of others, included information from in vitro transport assays, in vivo murine knockout studies, and human GWAS. These data support the importance of SLC22 genes in regulating levels of metabolites, signaling molecules and antioxidants in both cells and plasma. In the context of their altered expression in kidney cancer and their relationship to disease progression, this suggests that they may play a key role in tumor biology related to pathological stage. Moreover, their role in uptake of chemotherapeutic agents like methotrexate and cis-platinum raises the possibility that their altered expression may play a role in tumor responsiveness to certain chemotherapeutics.

In a subsequent. analysis, we identified ADME and related genes that were differentially expressed and generated interaction networks to visualize the SLC22 genes within a larger gene set (Figure 4A and Appendix A). Strong support for our hypothesis that the SLC22 genes are important was shown in their centrality to these networks as measured by degree and closeness centrality (Appendix A). Additional non-SLC22 genes, ABCG2 and SLC47A1 were also observed to be central to these networks (Figure 4B and Appendix A). The ABC transporter gene, ABCG2, transports urate in normal kidney tissue [68]. It has been shown to transport xenobiotics and play a role in multi-drug resistance to chemotherapeutic drugs in other types of cancer [69]. We observed significant changes in ABCG2 gene expression within multiple comparisons of the TNM classes of oncologic variables (Figure 5A–C) and also a larger association of expression with overall survival (Figure 6A). Previous work has also shown the relationship between ABCG2 expression and overall survival in KIRC [41,70]. Similarly, SLC47A1 was DE in multiple comparisons (Figure 5A–D) and shown previously to be significantly associated with overall survival in KIRC (Figure 6B) [71]. These results are strong confirmation that the further investigation of the central genes identified in this network approach are likely to have importance in the progression of KIRC and KIRP.

It is worth considering some strengths and limitations of what has been presented here. The TCGA dataset used is an invaluable resource due to the size of the cohort and the metadata available for each patient sample. However, these bulk RNA-Seq expression levels are many years old and newer technologies, such as single cell RNA-Seq, could provide better specificity and resolution regarding whether the genes studied here are decreased in both tumor and non-tumor cells. Similarly, other types of data like somatic mutation and protein abundance levels could provide additional context and support for the observed expression levels in the RNA-Seq data. Alternatively, the results described here provide some promising leads for diagnosis/prognosis of longer compared to shorter overall survival durations based on SLC22 gene expression. Further testing and validation are needed.

## 5. Conclusions

Using publicly available datasets from the TCGA project, we investigated the role of SLC22 transporters in the two most common types of renal cell carcinoma, KIRC and KIRP. Many important SLC22 genes—including those of the OAT and OAT-related groups—had decreased expression over the continuum of stages of renal cell carcinoma from well-functioning, healthy kidneys to advanced metastatic disease. This relationship is manifested, when accounting for the patient-specific demographics of gender and age at diagnosis, in the significant association of decreased SLC22 gene expression with reduced duration of overall survival. Most often, this association was deepened by accounting for reported race, but not factors that define pathologic disease progression. Alternatively, analysis of patients with different classifications of tumor size/progression, lymph node involvement, and presence of metastasis identified multiple SLC22 transporters as significantly changed, often decreasing with severity. Within the context of a network of transporter and metabolic genes, SLC22 genes are central along with other genes like ABCG2 and SLC47A1; and identifying the meaning of this centrality is likely to be an important step in understanding the stratification of overall survival within the population of kidney cancer patients. A number of the identified transporters (e.g., SLC22A12/URAT1, SLC22A6/OAT1, SLC22A8/OAT3, ABCG2) are well-established uric acid transporters; this may be clinically important since, as discussed above, a number of studies indicate that altered uric acid levels and kidney cancer are associated. According to the Remote Sensing and Signaling Theory, uric acid and other antioxidants, metabolites, and signaling molecules are transported by SLC22 and other transporters to regulate small molecule homeostasis across organelles, cells, organs, and organisms [1,6,38,39]. Concepts underlying this theory could be highly relevant to cancer biology. Moreover, these genes have the potential to be used as targets of treatment and/or biomarkers for overall survival, especially in advanced disease. There are few treatments for metastatic renal carcinoma, so the prospect of new potential therapeutic targets requires novel interpretations and analyses of the currently available data. In addition, our studies suggest that strategies for kidney cancer treatment need to consider the potential biomarkers found here. It is at least conceivable that present strategies may insufficiently treat disease in patients with low SLC22 expression but only localized disease. In contrast, the possibility exists that patients with advanced disease and higher SLC22 expression might be overtreated. It is important that future clinical studies address these possibilities.

## Figures and Tables

**Figure 1 cancers-14-04772-f001:**
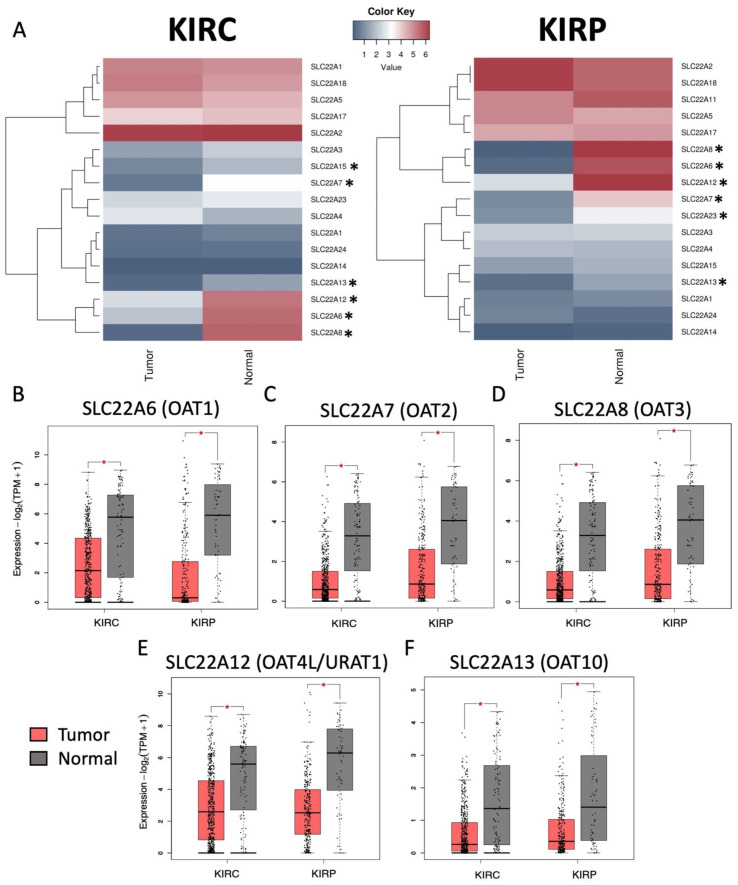
SLC22 gene expression changes between normal kidney tissue and KIRC and KIRP tumors. (**A**) Heatmaps generated by RNA-Seq data collected from the GEPIA tool showing all expressed SLC22 genes in KIRC (left; num(Tumor) = 523; num(Normal) = 100) and KIRP (right; num(Tumor) = 286; num(Normal) = 60) tumors or normal kidney based on average transcripts per million reads (TPM). Tumor sample data is based on the TCGA cohort and normal kidney samples include matched tissue from TCGA and the GTEX cohorts. Asterisks indicate differentially expressed (DE) genes between tumor and normal with 5 out of 7 DE genes shared between the tumor comparisons. Expression boxplots generated using the GEPIA2 tool compare the expression distribution (with individual points representing patient samples) in primary KIRC and KIRP tumors and matched normal kidney tissue and GTEX samples for SLC22A6 (**B**), SLC22A7 (**C**), SLC22A8 (**D**), SLC22A12 (**E**), SLC22A13 (**F**). For each of these genes in each comparison, the expression is significantly higher in normal samples compared to tumor based on limma differential gene expression analysis. Points represent log-transformed transcripts per million reads (TPM). *—limma adjusted *p*-value < 0.01.

**Figure 2 cancers-14-04772-f002:**
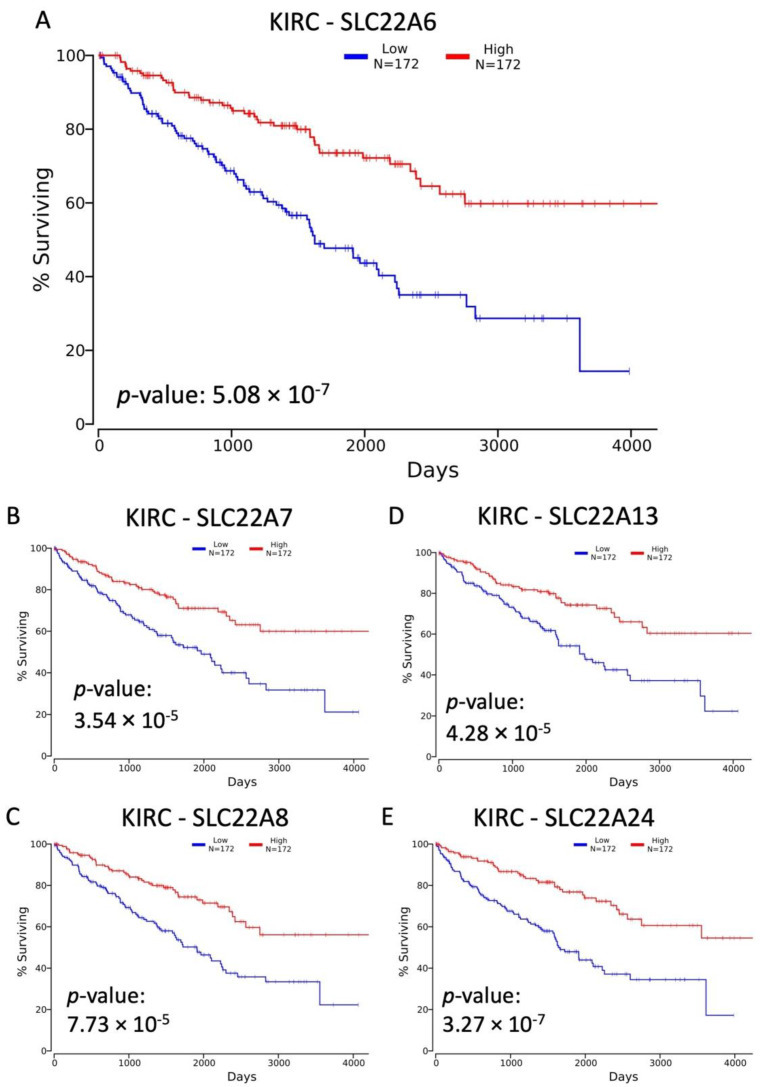
Kaplan–Meier plots for selected SLC22 genes in KIRC. Plot of the results of Kaplan–Meier survival analysis and logrank test showing the association of SLC22A6 (**A**), SLC22A7 (**B**), SLC22A8 (**C**), SLC22A13 (**D**), and SLC22A24 (**E**) expression and overall survival (in days) in KIRC. High expression is defined as the top tercile (n = 172) of patient tumor values and low expression represents the bottom tercile. The *p*-value of the logrank test is significant at less than 0.05.

**Figure 3 cancers-14-04772-f003:**
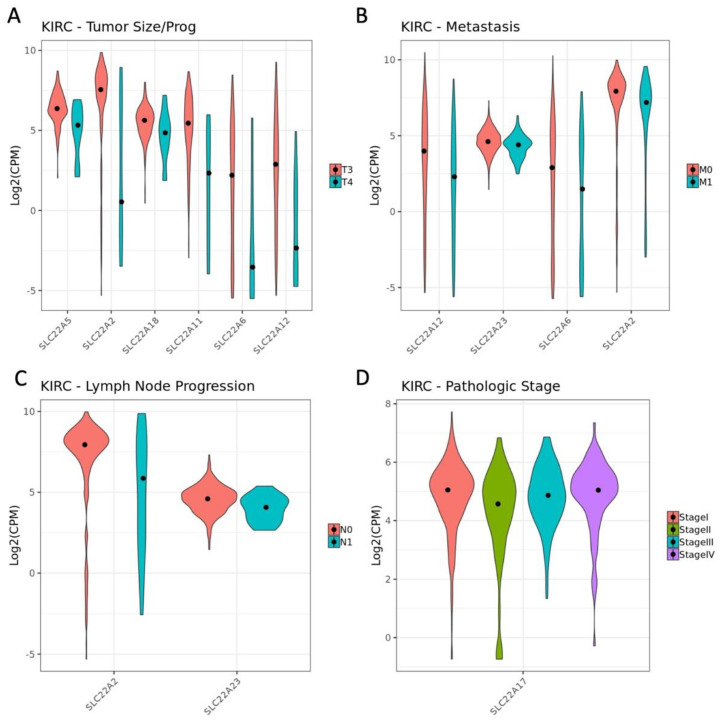
SLC22 gene expression by class for pathologic variables in KIRC. (**A**) Violin plot of expression of SLC22 genes differentially expressed (DE) in KIRC between two successive sample subsets of the variable Tumor Size and Progression (T3 and T4). Plots of expression of DE SLC22 genes in the categories for the variable Presence of Metastasis, M0 and M1 (**B**); Lymph Node Involvement, N0 and N1 (**C**); and Pathologic Stage, Stage I, Stage II, Stage III, Stage IV (**D**). Expression is measured as the log of counts per million (Log(CPM)).

**Figure 4 cancers-14-04772-f004:**
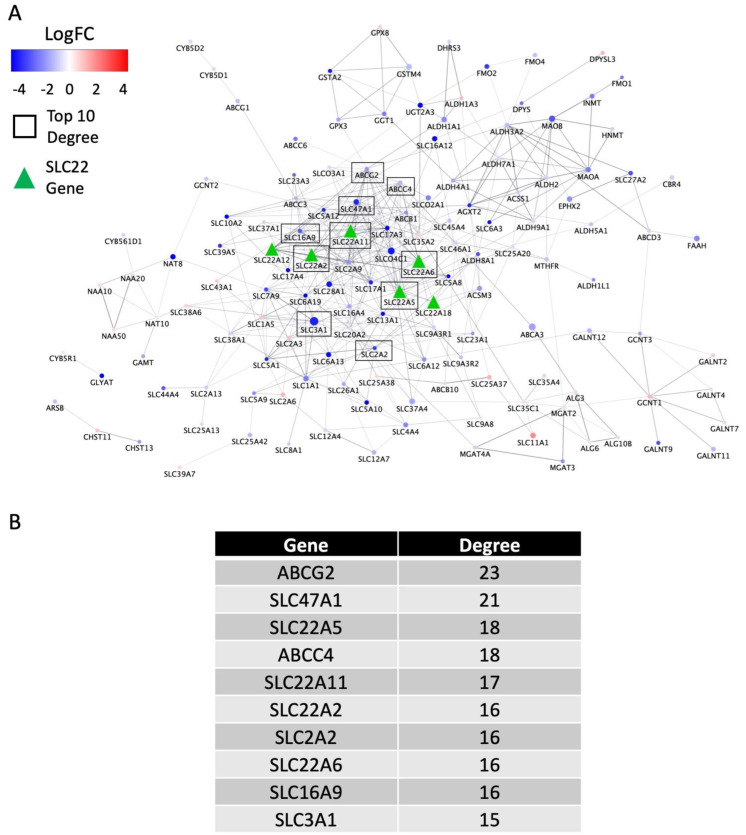
SLC22 gene network and metrics for tumor size/progression class 3 vs. 4 comparison. SLC22 and ADME interaction network composed of DE genes for the comparison of Tumor Size and Progression, T3 vs. T4 (**A**). The 424 Edges represent interactions found in the STRING interaction database and the transparency of the edges is a function of the Combined Score which is also stored in the database. With the exception of SLC22 genes (green triangles), nodes are colored by log fold-change in the relevant comparison and the node sizes correspond to the negative log(adjusted *p*-value). Nodes with the highest (top 10) degrees are identified within black boxes. (**B**) Table of the top 10 genes by degree identified by black boxes in (**A**).

**Figure 5 cancers-14-04772-f005:**
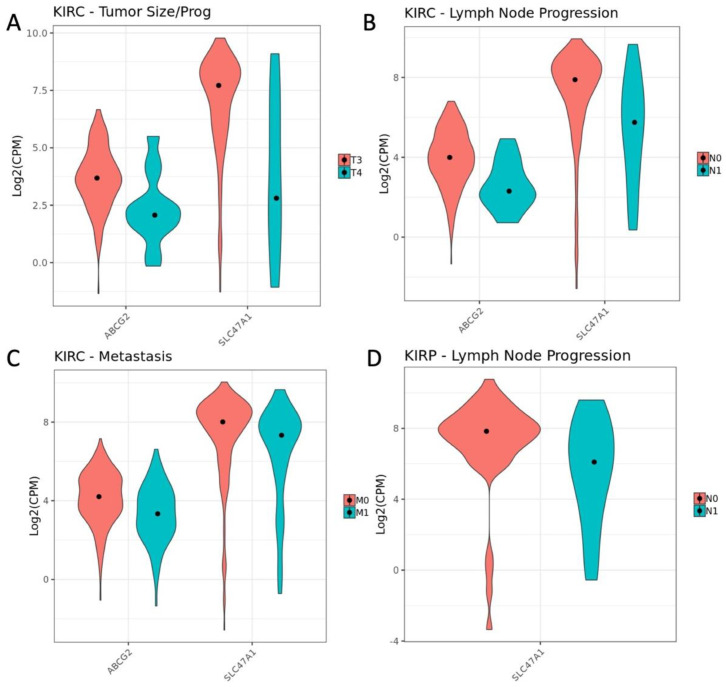
SLC22 interacting gene expression by class for pathologic variables in KIRC. Violin plot of expression of ABCG2 and SLC47A1 that are DE in KIRC between two successive sample subsets of the oncologic variable Tumor Size and Progression, T3 v T4 (**A**), Lymph Node Progression, N0 v N1 (**B**), and presence of Metastasis, M0 v M1 (**C**). Expression of SLC47A1 in KIRP Lymph Node Progression, N0 v N1 (**D**). Expression is measured as the log of counts per million (Log2(CPM)).

**Figure 6 cancers-14-04772-f006:**
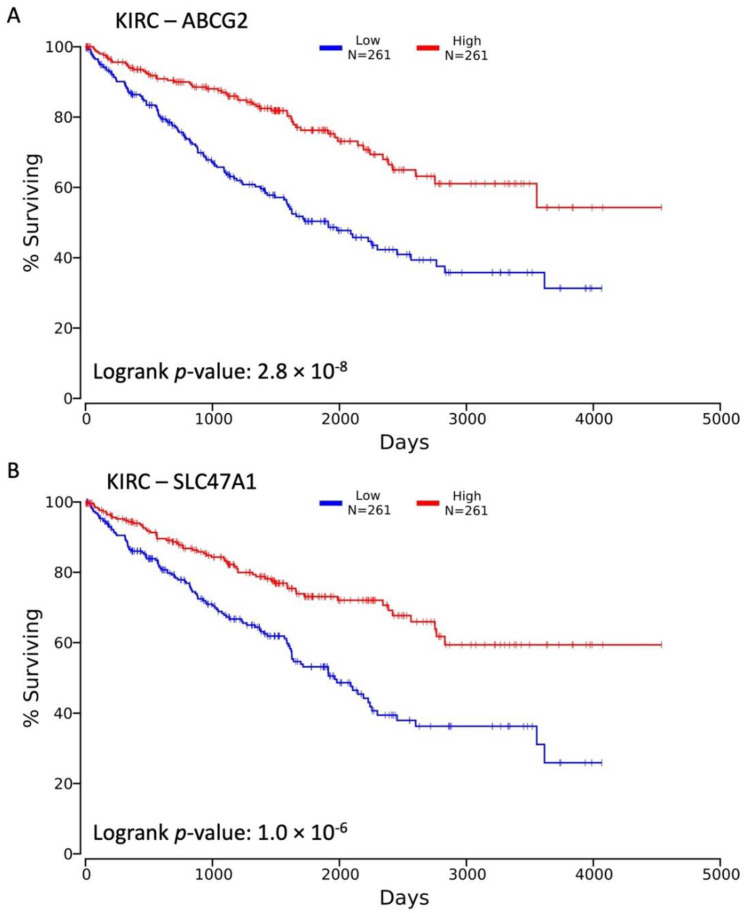
Kaplan–Meier plots for selected SLC22 interacting genes in KIRC. Association of ABCG2 and SLC47A1 and overall survival in KIRC is significant using Cox proportional hazards regression model and the Cox coefficient is negative indicating that higher expression is observed in patients with longer survival. This relationship is shown by Kaplan–Meier plot in KIRC for ABCG2 (**A**) and SLC47A1 (**B**). The red line represents patients with gene expression above the median for the gene of interest; the blue line represents patients with gene expression below the median. The *p*-values associated with logrank tests comparing the halves are also shown.

**Table 1 cancers-14-04772-t001:** SLC22 gene optimal Cox Proportional Hazards model results.

Symbol	EG ID	Cancer Type	Optimal Model *	KIRC Cox Coef.	Hazard Ratio
SLC22A2	6582	KIRC	Base + race + treatment type	−0.433	0.649
SLC22A4	6583	KIRC	Base + race + treatment type	−0.332	0.717
SLC22A5	6584	KIRC	Base + stage	−0.354	0.702
SLC22A6	9356	KIRC	Base + race + treatment type	−0.413	0.661
SLC22A7	10864	KIRC	Base + race + prior malignancy	−0.319	0.727
SLC22A8	9376	KIRC	Base + race + treatment type	−0.465	0.628
SLC22A11	55867	KIRC	Base + race + treatment type	−0.384	0.681
SLC22A12	116085	KIRC	Base + race + treatment type	−0.389	0.678
SLC22A24	283238	KIRC	Base + race + treatment type	−0.442	0.643
SLC22A2	6582	KIRP	Base + prior malignancy	−0.911	0.631
SLC22A13	9390	KIRP	Base + race + lymph node involvement	−0.668	0.513
SLC22A18	5002	KIRP	Base + race + tumor size/prog	−0.644	0.525
SLC22A24	283238	KIRP	Base + race + stage	−0.757	0.469

* Base model: SLC22 gene expression + age + gender.

**Table 2 cancers-14-04772-t002:** SLC22 gene annotation and OncoLnc survival analysis results.

Sym	Alias	EG ID	Subclade *	Updated Grouping **	Specificity	KIRC Cox Coef.	KIRC BH-adj *p*-val	KIRP Cox Coef.	KIRP BH-adj *p*-val
SLC22A1	OCT1	6580	OCT	OCT	Multi-	**0.2069**	^&^ **0.02783**	0.0921	0.69794
SLC22A2	OCT2	6582	OCT	OCT	Multi-	−0.3411	^+&^ 0.00033	**−0.9214**	**6.94** **× 10^−5^**
SLC22A3	OCT3	6581	OCT	OCT	Oligo-	0.0074	0.94686	−0.3733	0.08201
SLC22A4	OCTN1	6583	OCTN	OCTN- related	Oligo-	**−0.3159**	**0.00029**	−0.3289	0.16737
SLC22A5	OCTN2	6584	OCTN	OCTN- related	Oligo-	**−0.2901**	^#^ **0.00281**	−0.4169	0.07343
SLC22A6	OAT1	9356	OAT	OATS1	Multi-	**−0.3473**	^+^ **0.00015**	−0.0640	0.82770
SLC22A7	OAT2	10864	OAT	OATS2	Oligo-	**−0.2599**	**0.00814**	−0.3251	0.15414
SLC22A8	OAT3	9376	OAT	OATS1	Multi-	**−0.3883**	**9.46** **× 10^−5^**	NA	NA
SLC22A11	OAT4	55867	OAT	OATS3	Oligo-	**−0.3384**	**0.00049**	−0.3131	0.12484
SLC22A12	URAT1	116085	OAT	OATS3	Mono-	**−0.3211**	^ **0.00135**	−0.3657	0.07921
SLC22A13	ORCTL3	9390	OAT-like	OAT-like	Mono-	**−0.2307**	**0.02281**	**−0.4095**	**0.04680**
SLC22A14	ORCTL4	9389	OAT-like	OAT-like	N/A	**0.2288**	**0.01846**	−0.2331	0.32010
SLC22A15	FLIPT1	55356	OCTN- related	OCTN-related	Mono-	0.1598	0.08677	0.2117	0.33305
SLC22A17	BOCT	51310	OAT- related	OAT-related	Mono-	0.0689	0.54035	−0.2479	0.23082
SLC22A18	ORCTL2	5002	OAT- related	OAT-related	N/A	−0.0784	0.47059	**−0.4283**	**0.04613**
SLC22A23	N/A	63027	OAT- related	OAT-related	Oligo-	−0.0992	0.35122	0.4132	0.05213
SLC22A24	N/A	283238	OAT	OATS4	Oligo-	**−0.3146**	**0.00200**	**−0.5923**	**0.00669**

* Defined in Zhu et al., 2015 [16]; ** Defined in Engelhart et al., 2020 [17]; and ^&^ shown previously in Ciarimboli et al., 2020 [40]; ^+^ Shown previously in Hu et al., 2020 [41]; ^#^ Shown previously in Edemir et al., 2020 [42]; ^ Shown previously in Xu et al., 2021 [43].

**Table 3 cancers-14-04772-t003:** Summary of pathologic variable DE analysis in KIRP and KIRC.

Oncologic Variable	Comparison	KIRP-SLC22 (*p* adj. < 0.1)	KIRC-SLC22 (*p* adj. < 0.1)
Stage	Stage I v Stage II	0	**2**
	Stage II v Stage III	0	**1**
	Stage III v Stage IV	0	0
T	T1 v T2	0	0
	T2 v T3	0	0
	T3. v T4	0	**6**
M	M0 v M1	0	**4**
N	N0 v N1	**5**	**2**
PriMalig	Yes v No	0	0
TxType	Pharm v Rad	0	0

## Data Availability

Publicly available datasets were analyzed in this study. TCGA (The Cancer Genome Atlas) data were accessed using the TCGA portal (http://portal.gdc.cancer.gov, accessed on 24 March 2021) to download RSEM normalized gene expression files and patient clinical metdata for KIRC and KIRP. The differential gene expression analysis and visualization comparing KIRC and KIRP tumors with matched normal tissue was conducted in GEPIA2 (http://gepia2.cancer-pku.cn/, accessed on 19 May 2021). The primary survival analysis and cox proportional hazards regression was conducted with the OncoLnc tool (www.oncolnc.org, accessed on 3 May 2021, [35]). Network analysis was conducted with the STRING tools (www.string-db.org, accessed on 19 April 2022). All other analyses were performed in the R environment (v3.5.1). The data and code presented in this study are available at https://github.com/ucsd-ccbb/SLC22_TCGA_Methods (accessed on 17 August 2022).

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
