# Peer review of "Organic Anion Transporters (OAT) and Other SLC22 Transporters in Progression of Renal Cell Carcinoma"

_cancers, 2022, doi:10.3390/cancers14194772_

Round 1

Reviewer 1 Report

This study statistically analyzes the connection between various members of the solute-linked carrier family 22 (SLC22) genes and KIRC/KIRP RNA sequencing datasets available on-line from the TCGA project. The study objective is undoubtedly of interest to oncologists, because until now the role of SLC22 and other SLC genes was insufficiently investigated regarding their involvement in the cancerogenesis and tumor patients’ survival.

Recent literature increasingly suggests that poor patient outcome is linked to an aberrant SLC expression. Therefore this report is likely to be of particular interest to molecular oncologists and cancer researchers in general.

Although the authors present a sound study, there are some points that need to be addressed, given below:

General comment:

The Abstract should include a clear statement summarizing the main finding/achievement of the study.

Specific points

1). page 2, line 41: “Solute-like carrier (SLC) genes” ,
The acronym SLC stands for “Solute-Linked Carrier”

2). pp. 7-8

Lines 245, 252, 253:          “Figure S1A-E” should be corrected to  “Figure S2A-E”

3). p. 9

Line 296:      Table S4 appears to be redundant because it does not add any additional information already given in the main text.

Line 299:      “…Where there were 6…” – Please clarify this confusing sentence.
“6” what?

4). p. 10

Legend to Figure 3 lines 310-312: The sentence is worded in a confusing manner. Please rephrase it.

5). p. 11

Lines 314-315:        The authors refer to Fig. S4, which is about KIRC. But the sentence in line 314 says “In KIRP”. Which one is it?
Also in line 315, they write “genes with 4”. What does it mean?

Lines 317-318:        Again, the lines highlighted in orange in Table S5 refer to KIRC and not to KIRP as described in the text. Please correct.

Line 362:                 “Figure S4A” should be corrected to “Figure S5A”

6). p. 13 Figure 5:   Figure 5, its legend and the main text (lines 377-383) apparently do not match.

7). p. 14 Figure 6:   What do the authors mean with „in the higher/lower half”? Half of what? Please be more precise.

8. Supplement:

Table S5:      The title of Table S5 does not mention KIRP.

Figure S4:     KIRC or KIRP?

Author Response

Response to Reviewer #1

General comment:

The Abstract should include a clear statement summarizing the main finding/achievement of the study.

Response: We appreciate this comment and have revised the abstract to include a sentence highlighting the main take home message of the study. We included Background, Results, and Implications in the abstract for clarity.

Specific points:

1). page 2, line 41: “Solute-like carrier (SLC) genes” ,

The acronym SLC stands for “Solute-Linked Carrier”

Response: We double checked multiple reviews and the field generally accepts the term “SoLute Carrier“ for SLC (i.e. PMID: 21186191; PMID: 19164095).

2). pp. 7-8

Lines 245, 252, 253:          “Figure S1A-E” should be corrected to  “Figure S2A-E” 

Response: Due to the article format of the template, the methods were placed ahead of the results. The first supplemental figure referenced in the text is found in the Methods which is the reason that the first supplemental figure referenced in the Results is labeled S2.

3). p. 9

Line 296:      Table S4 appears to be redundant because it does not add any additional information already given in the main text.

Response: We agree it is redundant with the main text and have removed it.

Line 299:      “…Where there were 6…” – Please clarify this confusing sentence. “6” what?

Response: We apologize for the confusing sentence. It has been updated to read, “While multiple comparisons had no DE genes in the entire expressed genelist (i.e. Stage III vs Stage IV), there were 6 DE SLC22 genes in the Tumor Size and Progression comparison in KIRC [SLC22A2, SLC22A5, SLC22A6, SLC22A11, SLC22A12, SLC22A18]”. 

4). p. 10

Legend to Figure 3 lines 310-312: The sentence is worded in a confusing manner. Please rephrase it.

Response: We have updated this legend to read, “(A) Violin plot of expression of SLC22 genes DE in KIRC between two successive sample subsets of the variable Tumor Size and Progression (T3 and T4). Plots of expression of DE SLC22 genes in the categories for the variable Presence of Metastasis, M0 and M1 (B); Lymph Node Involvement, N0 and N1 (C); and Pathologic Stage, Stage I, Stage II, Stage III, Stage IV (D)”.

5). p. 11

Lines 314-315:        The authors refer to Fig. S4, which is about KIRC. But the sentence in line 314 says “In KIRP”. Which one is it? Also in line 315, they write “genes with 4”. What does it mean?

Response: We have updated the figure title, plot title, and main text to reflect the changes in KIRP which are discussed in this paragraph. The confusing sentence has been updated to read, “In KIRP, only positive versus negative lymph node involvement (N0 vs N1) showed any DE SLC22 genes which were SLC22A2, SLC22A18, SLC22A3, SLC22A11, and SLC22A5 (Figure S4).”.

Lines 317-318:        Again, the lines highlighted in orange in Table S5 refer to KIRC and not to KIRP as described in the text. Please correct.

Response: We have updated the title for Table S4, which was previously Table S5, to include KIRP. In reviewing the text, we found the sentence referring to this table to be unclear about what was highlighted in the table. We have updated the text which is meant to show that genes that are DE in the comparisons of oncologic variable subgroups (i.e. N0 vs N1) overlap with DE genes between Normal and Tumor or with genes associated with overall survival.

Line 362:                 “Figure S4A” should be corrected to “Figure S5A”

Response: We have updated the main text to correct this error. 

6). p. 13 Figure 5:   Figure 5, its legend and the main text (lines 377-383) apparently do not match.

Response: We have corrected the error that mentioned ABCG2 as DE in panel D of Figure 5 and replaced it with SLC47A1 which is shown in the figure and mentioned in the figure legend.

7). p. 14 Figure 6:   What do the authors mean with „in the higher/lower half”? Half of what? Please be more precise.

Response: We apologize for the vague description. For this analysis, patients are classified as low or high based on whether their gene expression level for ABCG2 or SLC47A1 is below or above the median of all patients. We have updated the figure legend to clarify this point.

Supplement:

Table S5:      The title of Table S5 does not mention KIRP.

Response: As mentioned above for the text referring to Table S4, which was previously Table S5, we have added KIRP to the title. 

Figure S4:     KIRC or KIRP?

Response: As mentioned above for the text referring to Figure S4, we have updated the figure title and plot title to correct this error.

Reviewer 2 Report

The authors explore the role of SLC22 transporters in clear cell renal carcinoma (KIRC) and papillary cell renal carcinoma (KIRP). Topic of moderate interest. Good scientific accuracy and general organization of the paper. Good use of English.

Some major issues are present together with several minor issues.

Although the abstract should be unstructured for editorial rules, I recommend that authors sequentially include background, objectives, methods, results and conclusions to facilitate reading and understanding. Therefore, the authors should keep abstract unstructured but make it complete.

The authors should report the meaning of all acronyms when used the first time in Abstract, Full-text and Legends of tables/figures

There are several risk factors for RCC, not only obesity and smoking. I suggest to include also hypertension, familiarity, end-stage renal disease, chemical carcinogens. Besides, this is a useful reference 10.1016/j.clgc.2018.09.010

Lines 97-103: this part should be moved in the Methods and Results section.

The author in the Methods should clearly define the endpoints (primary and secondaries).

Why the authors used different colors in Table 1 and 2? The authors should clarify this point

I suggest to add a brief section with Strengths and Limitations of study. Besides, I suggest emphasizing the future prospects of research more

Author Response

Response to Reviewer #2

Although the abstract should be unstructured for editorial rules, I recommend that authors sequentially include background, objectives, methods, results and conclusions to facilitate reading and understanding. Therefore, the authors should keep abstract unstructured but make it complete.

Response: Thank you for the suggestion. We have revised the abstract to follow the recommended order.

The authors should report the meaning of all acronyms when used the first time in Abstract, Full-text and Legends of tables/figures

Response: We understand the reviewer’s comment and have made the appropriate changes to the Full-text and table/figure legends.

There are several risk factors for RCC, not only obesity and smoking. I suggest to include also hypertension, familiarity, end-stage renal disease, chemical carcinogens. Besides, this is a useful reference 10.1016/j.clgc.2018.09.010

Response: We appreciate the insight into additional risk factors provided by the reviewer. We have updated the text to reflect these additional factors and included the reference which provided useful input for the discussion paragraph on survival.

 The author in the Methods should clearly define the endpoints (primary and secondaries). 

Response: We have added a sentence in the Methods to describe the primary endpoint based on what was available from the TCGA dataset.

 Why the authors used different colors in Table 1 and 2? The authors should clarify this point

Response: Due to confusion mentioned by multiple reviewers, we have removed the use of colors from Tables 1 and 2. Instead we use bold text to highlight which genes were significantly associated with overall survival in each type of renal cell carcinoma. 

 I suggest to add a brief section with Strengths and Limitations of study. Besides, I suggest emphasizing the future prospects of research more

Response: We thank the reviewer for this suggestion and have added a paragraph to the discussion briefly describing both strengths and limitations. We have also included some additional sentences in the abstract and conclusions emphasizing the potential value of this research in the future.

Reviewer 3 Report

In this manuscript, Whisenant and Nigam investigated datasets from TCGA to examine the role of SLC22 transporters in the two most common types of renal cell carcinoma, KIRC and KIRP.

The authors assess the relationship between SLC22 gene expression and survival in patients with kidney cancer and identify that low expression of several of these genes have a significant impact on disease progression and overall patient survivorship. Due to the poor prognosis for patients diagnosed with metastatic renal cell carcinoma, this paper utilizes publicly available data on the disease to identify biomarkers to provide more personalized treatment.

Compliments: The authors considered how several factors such as age, race, and sex could impact how SLC22 expression impacts disease progression and survivorship. The authors highlight the importance of identifying biomarkers for this disease to help change the current standard of care when it comes to KIRC and KIRP patients. Current criteria that are being used to identify the course of treatment does not take into account these potential biomarkers. With this current method there is a high probability that we are undertreating the disease for patients with low SLC22 expression with only localized disease. This alternatively could result in overtreating the more advanced disease diagnoses with patients who have higher SLC22 expression.

Overall, the authors present a well-formulated article with easy-to-understand data presentation. However, the paper could be improved by enhancing some of the figures/tables and some grammar as per my recommendation.

MAJOR ISSUES

1) TCGA clinical metadata variable “AGE” was added to the models used in DE comparisons, however, no analysis was included on patient age upon disease diagnosis/onset and SLC22 expression levels. Was this analysis performed? Did low SLC22 expression correlate with younger age at time of diagnosis?

No other major issues requiring further experimental analysis were identified in this paper.

MINOR ISSUES

1) Network analysis in methods section. STRING database was used to identify any protein-protein-interactions with SLC22 that were later mapped out using Cytoscape. Were any exclusion criteria used on the website for the confidence in the interactions included in Figure 4a? Please identify these in this section.

2) Table 1 and 2. Please define the text colours directly under the tables.

3) Lines 187-189. “Of the human SLC22 transporters of organic cations, anions, and zwitterions, there 187 are 17 expressed at detectable levels in KIRC and KIRP based onthe publicly available 188 RNA-Seq data…”

The author mentions that there are 17 transporters detected, perhaps adding in brackets the proportion of each type could be beneficial to the reader. (e.g., cations [8], anions [6], and zwitterions [3].)

4) Lines 248 – 249. “…it is worth noting that all of them 248 are strongly implicated in transport of the anti-oxidant, uric acid.”

The reader can benefit by understanding the implications of impaired uric acid uptake in renal cells here. Literature indicates that Serum Uric Acid (SUA) correlates with incidence of renal cancer. Please cite accordingly.

5) Lines 276-277. “… there were 9 of 12 (Figure S3A) and 4 of 4 (Figure S3B) SLC22 genes 276 still significantly associated with overall survival in KIRC and KIRP, respectively”

Please include the 9 and 4 gene names in brackets respectively.

6) On line 279 the authors indicate that a separate analysis was needed to determine the relationship between the added variables and SLC22 expression. Please specify what this analysis was.

7) On line 300 you mention 4 DE SLC22 genes were identified for the presence of metastatic disease, please list these in brackets. Please do this for the other genes that were only identified by number of genes in this section.

ON PRESENTATION AND STYLE

1) Please correct all “--"t; to “–“.

2) Figure 1A. Asterisk should be better aligned with gene name to ensure the audience isn’t perceiving the wrong gene being identified.

3) The authors interchangeably refer to the genes in both aliases. After identifying (e.g., SLC22A7 as OAT2) please choose only one naming throughout. Perhaps an additional column could be added to Table 1 or 2 clearly showing the alias for each.

Author Response

Response to Reviewer #3

The authors considered how several factors such as age, race, and sex could impact how SLC22 expression impacts disease progression and survivorship. The authors highlight the importance of identifying biomarkers for this disease to help change the current standard of care when it comes to KIRC and KIRP patients. Current criteria that are being used to identify the course of treatment does not take into account these potential biomarkers. With this current method there is a high probability that we are undertreating the disease for patients with low SLC22 expression with only localized disease. This alternatively could result in overtreating the more advanced disease diagnoses with patients who have higher SLC22 expression.

 Response: We thank the reviewer for these comments and we have further emphasized the need to clinically evaluate this issue of treating based on SLC22 expression levels.

Overall, the authors present a well-formulated article with easy-to-understand data presentation. However, the paper could be improved by enhancing some of the figures/tables and some grammar as per my recommendation.

MAJOR ISSUES

1) TCGA clinical metadata variable “AGE” was added to the models used in DE comparisons, however, no analysis was included on patient age upon disease diagnosis/onset and SLC22 expression levels. Was this analysis performed? Did low SLC22 expression correlate with younger age at time of diagnosis?

Response: We understand the reviewer’s concern with using age in the models for the DE comparisons. The use of age (and gender) in the limma models was based on their inclusion in the cox proportional hazards models and because these variables were used by the GEPIA tool in their limma analysis. As suggested by the reviewer, we have performed the correlation analysis between age (upon disease diagnosis) and SLC22 gene expression and found no significant correlations. We have stated in the Methods that age was significantly associated with overall survival in cox proportional hazards models without the SLC22 gene expression used as a variable. This suggests that, by including age in the model, we are removing some biological variation related to survival. But given the absence of correlation between age and SLC22 gene expression, we assume there is not an increased likelihood of identifying differentially expressed SLC22 genes when the age variable is included in the limma models.

We have added additional text to explain this rationale, “We determined there is no direct correlation between SLC22 gene expression and age or gender in the TCGA renal cancer data [data not shown]. In addition, there is a precedent for using these variables in linear models to obtain lists of differentially expressed genes [30]. Furthermore, in two separate Cox proportional hazards models in KIRC that were independent of gene expression, age was significantly associated with overall survival (Figure S1A-B). These results suggest that any biological variation associated with age will be removed in the limma models and increase our confidence in any significant DE SLC22 genes.”

MINOR ISSUES

1) Network analysis in methods section. STRING database was used to identify any protein-protein-interactions with SLC22 that were later mapped out using Cytoscape. Were any exclusion criteria used on the website for the confidence in the interactions included in Figure 4a? Please identify these in this section.

 Response: In the Methods, it is stated that we used STRING interactions with Confidence Score > 0.4. We have added additional text to state that interactions with score less than 0.4 have been excluded.

2) Table 1 and 2. Please define the text colours directly under the tables.

 Response: Due to confusion mentioned by multiple reviewers, we have removed the use of colors from Tables 1 and 2. Instead we use bold text to highlight which genes were significantly associated with overall survival in each type of renal cell carcinoma.

3) Lines 187-189. “Of the human SLC22 transporters of organic cations, anions, and zwitterions, there 187 are 17 expressed at detectable levels in KIRC and KIRP based onthe publicly available 188 RNA-Seq data…”

The author mentions that there are 17 transporters detected, perhaps adding in brackets the proportion of each type could be beneficial to the reader. (e.g., cations [8], anions [6], and zwitterions [3].) 

Response: We thank the reviewer for the suggestion and we have explained these relationships in the text corresponding to Table 2.

4) Lines 248 – 249. “…it is worth noting that all of them 248 are strongly implicated in transport of the anti-oxidant, uric acid.”

The reader can benefit by understanding the implications of impaired uric acid uptake in renal cells here. Literature indicates that Serum Uric Acid (SUA) correlates with incidence of renal cancer. Please cite accordingly.

Response: We appreciate the reviewer pointing out this relevant literature. We have added a couple of sentences to put this finding in context and made the relevant citations.

5) Lines 276-277. “… there were 9 of 12 (Figure S3A) and 4 of 4 (Figure S3B) SLC22 genes 276 still significantly associated with overall survival in KIRC and KIRP, respectively”

Please include the 9 and 4 gene names in brackets respectively.

Response: We have updated the text to reflect this suggestion.

6) On line 279 the authors indicate that a separate analysis was needed to determine the relationship between the added variables and SLC22 expression. Please specify what this analysis was.

Response: We understand that this sentence was unclear. The sentence was meant as a transition from the overall survival analysis subsection to the differential expression analysis subsection. The text has been updated to clearly state this transition. “From these results, we determined that the oncologic variables tested were not part of highest performing proportional hazards models and a separate analysis utilizing these variables in limma models for differential expression is required to determine their relationship to SLC22 expression.” 

7) On line 300 you mention 4 DE SLC22 genes were identified for the presence of metastatic disease, please list these in brackets. Please do this for the other genes that were only identified by number of genes in this section.

Response: We have updated the text to reflect this suggestion.

ON PRESENTATION AND STYLE 

1) Please correct all “--"t; to “–“.

2) Figure 1A. Asterisk should be better aligned with gene name to ensure the audience isn’t perceiving the wrong gene being identified.

3) The authors interchangeably refer to the genes in both aliases. After identifying (e.g., SLC22A7 as OAT2) please choose only one naming throughout. Perhaps an additional column could be added to Table 1 or 2 clearly showing the alias for each.

Response: We have made these suggested corrections throughout the manuscript. In addition, we have added an alias column to Table 2.

Round 2

Reviewer 2 Report

The authors changed the paper according to my suggestions